# TIME-SERIES BASED QUANTUM STATE DISCRIMINATION

## ABSTRACT

Measurement errors in quantum computers are very detrimental to quantum computations. The ability to efficiently and accurately readout quantum states is crucial for quantum error correction schemes and quantum algorithms. Readout fidelity is typically limited by a poor signal-to-noise (SNR) ratio between the quantum states we intend to classify, as well as energy relaxation (e.g., $T_1$ decay) from an excited state to a lower state during readout. Superconducting quantum bits (qubit), one of the leading candidates for scalable quantum computing hardware, are particularly limited by energy relaxation due to their relatively short coherence times. While most approaches for classifying the results of readout on superconducting qubits typically utilize clustering algorithms (e.g., a Gaussian mixture model) on integrated readout signals, these cannot distinguish between a quantum bit that was in the ground state prior to measurement from a qubit that decays to the ground state during measurement. For this reason, we instead propose using machine learning (ML) on the raw (non-integrated) analog signal and classification models on the full time series data (i.e., the *trajectory*). We observe that time series classification methods, such as our chosen long short-term memory (LSTM) model, in combination with filtering and feature engineering techniques, consistently outperform clustering models. In particular, we find that the largest improvements come from reclassifying points in the boundary regions between neighboring clusters. These boundary points correspond to measurement records that deviate from the typical cluster, likely due to transient or noisy features in the signal that are not captured when the data is integrated. By retaining temporal information, sequence-aware models such as LSTMs can better discriminate these trajectories, whereas clustering methods based on integrated values are more prone to misclassifications.

## 1 INTRODUCTION

Quantum computing offers exponential speedups for a range of important problems, including cryptography and simulation (Shor, 1997a; Arute et al., 2019; King et al., 2024). However, to outperform state-of-the-art classical supercomputers, quantum processors must achieve significantly lower error rates. This requirement can be met through error-correcting codes, as well as continual improvements in hardware, control, and qubit readout (Gidney and Ekerå, 2021; Shor, 1997b). Among leading platforms, superconducting qubits stand out for their scalability and fast gate operation times (Kjaergaard et al., 2020), but they are also highly susceptible to noise and decoherence (Devoret and Schoelkopf, 2013). In fact, recent experiments with a 142-qubit superconducting processor revealed that measurement and reset errors dominate the total error budget, underscoring the critical need for more accurate readout methods (Acharya et al., 2023).

Superconducting qubit readout is typically performed using dispersive coupling, where a resonator coupled to the qubit shifts its frequency depending on the qubit state (Wallraff, 2005). An analog probe pulse traverses the resonator and acquires state-dependent amplitude and phase shifts, which are digitized into in-phase (I) and quadrature (Q) components (Reed et al., 2010; Walter et al., 2017). After integrating these signals over the readout window, each measurement corresponds to a single point in the I-Q plane, forming clusters associated with different qubit states. The most widely used method for distinguishing these clusters is the Gaussian Mixture Model (GMM), which fits a two-dimensional Gaussian distribution to the integrated data (Jeffrey et al., 2014).

Beyond GMMs, supervised learning approaches such as feed-forward neural networks and support vector machines have also been explored (Magesan et al., 2015; Vora et al., 2024). While these integration-based methods provide relatively fast and high-fidelity readout, they often fail in regions where temporal information is essential, particularly for signals corrupted by stochastic noise, decay, or measurement-induced transitions (Khezri et al., 2023; Vijay et al., 2011; Gambetta et al., 2007). These limitations highlight the need for readout strategies that explicitly leverage the temporal structure of the signal.

In this work, we introduce a classification framework based on a long short-term memory (LSTM) network applied directly to the raw time-series data, combined with filtering and feature-engineering techniques. By preserving temporal correlations that are lost in integration-based schemes, our approach provides robustness against transient errors and is especially effective in correctly classifying states near cluster boundaries where decay events are most prevalent. We benchmark our method against conventional approaches using experimental data collected from superconducting qubits in a laboratory environment, demonstrating consistent improvements in classification accuracy.

The remainder of this paper is organized as follows. Section 2 provides an overview of the underlying physics of superconducting qubits. Section 3 describes the experimental setup and readout architecture, along with the classification approaches we evaluated. Section 4 presents a detailed comparison of classification performance, highlighting the regimes where our method offers significant gains. Finally, Section 5, 6 concludes with a discussion of related works and potential directions for future research.

## 2 BACKGROUND

### 2.1 QUANTUM BITS

Quantum bits, or qubits, encode the quantum states of a system into computational states. Unlike classical bits that can only take values of 0 or 1, qubits can exist in coherent superpositions of both states. This property is often visualized on the Bloch sphere (Figure. 1), where any point on the sphere corresponds to a valid qubit state. Extending this concept, our work considers *qutrits*, which allow three basis states $|0\rangle$, $|1\rangle$, and $|2\rangle$, with a general state given by

$$|\psi\rangle = a\,|0\rangle + b\,|1\rangle + c\,|2\rangle . \tag{1}$$

Such higher-dimensional quantum systems expand the available Hilbert space and enable richer encodings than classical counterparts, offering significant potential speedups in applications such as quantum simulation and cryptography (Grover, 1996; Shor, 1997a).

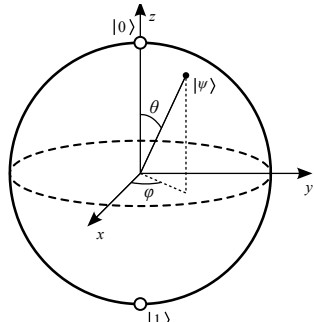

Figure 1: Bloch sphere representation.

Among various hardware platforms, superconducting transmon qubits are one of the most widely used due to their scalability and fast gate operation times (Kjaergaard et al., 2020). In these systems, computational states correspond to physical energy eigenstates of a Josephson junction circuit (Koch et al., 2007). However, superconducting qubits are inherently open quantum systems and are subject to energy relaxation and dephasing. The dominant error channel is energy relaxation, or $T_1$ decay, where an excited state stochastically relaxes to a lower state. Additional contributions from dephasing processes, collectively characterized as $T_2$ decay, further degrade coherence (Gambetta et al., 2007; Vijay et al., 2011). These decay mechanisms lead to measurement errors when a qubit partially relaxes during the short time of the readout pulse, making accurate state discrimination especially challenging.

To evaluate our proposed time-series classification methods across a range of experimental conditions, we collected readout data from eight fixed-frequency transmon qubits, with coherence times spanning from $24\,\mu s$ up to $120\,\mu s$ for the $|1\rangle$ state. These qubits serve as a representative testbed for assessing classification performance under realistic noise and decay dynamics in superconducting architectures.

## 2.2 READOUT SETUP

Our experiments were performed using the QubiC 2.0 control system, an FPGA-based platform for scalable qubit control and readout (Xu et al., 2023; 2021). As shown in Fig. 2, qubit drive and readout pulses are generated by an RFSoC (Radio Frequency System on Chip) with digital-to-analog converters (DACs), transmitted through the cryogenic stack to the qubit processor, and then amplified and digitized by analog-to-digital converters (ADCs).

In the standard approach, the readout signal is mixed with a local oscillator (DLO) and then integrated over the duration of the pulse to obtain a single I-Q point for classification. This integration process removes most temporal information, making methods such as Gaussian Mixture Models (GMMs) the conventional choice for state discrimination. In contrast, our setup records the non-integrated, mixed time-series directly from the ADC using an acquisition buffer. This preserves temporal dynamics in the readout signal, enabling sequence-aware models such as LSTMs to exploit information that would otherwise be averaged out.

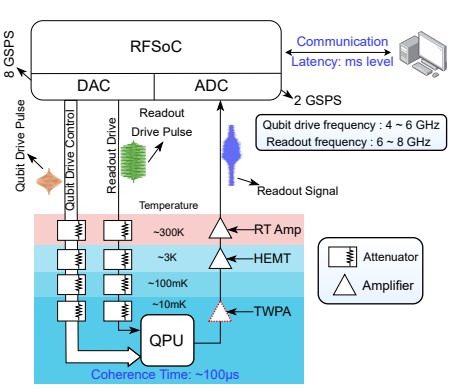

Figure 2: Control and readout schematic

To improve measurement fidelity, we implement a heralding step before each experiment to verify that the qubit is in the ground state $|0\rangle$, discarding shots that fail preparation (Johnson et al., 2012). We then insert a 1 $\mu$s delay between the heralding readout and the subsequent drive pulse to allow residual photons in the resonator to dissipate. Additionally, small state-dependent delays (120 ns for $|0\rangle$ and 60 ns for $|1\rangle$) are applied between the drive and readout pulses. These delays ensure that all circuits share a consistent effective start time, preventing artificial phase offsets that could bias time-series classification.

Each readout pulse lasted 750–1000 ns and was digitized at two samples per nanosecond, yielding 1500–2000 sequential timesteps per shot. This high-resolution dataset forms the basis for our comparison between integration-based classifiers and sequence-aware approaches.

## 3 APPROACH

### 3.1 DATA PRE-PROCESSING

Directly applying a time-series classifier to raw readout traces is not effective, as the signals are dominated by environmental and electronic noise. To improve classifier performance, we first applied pre-processing techniques to extract more informative features while suppressing spurious fluctuations. In this work, we focused on two complementary approaches: path-based feature engineering and bandpass filtering. The impact of these methods on the readout signal can be seen in Figures. 3 and 4.

#### 3.1.1 PATH FEATURES

Errors in qubit readout can arise from transient fluctuations, environmental noise, or relaxation processes that occur during the measurement window. To better capture these time-dependent effects, we engineered path-based features inspired by prior work (Cao et al., 2025). The path transform of a signal is defined as

$$X(t) = \int_0^t w(\tau)dX(\tau), \tag{2}$$

where each time step accumulates the weighted contributions of all previous samples. This cumulative representation reduces the impact of short-timescale noise while emphasizing longer-term

dynamics across the readout pulse. As illustrated in Figure. 4, raw trajectories exhibit irregular fluctuations, whereas the path-transformed trajectories form smoother and more separable curves between states. We additionally evaluated path signatures (Chevyrev and Kormilitzin, 2025), which provide a systematic way to compress two-dimensional trajectories into compact feature sets, yielding 63 features at order five.

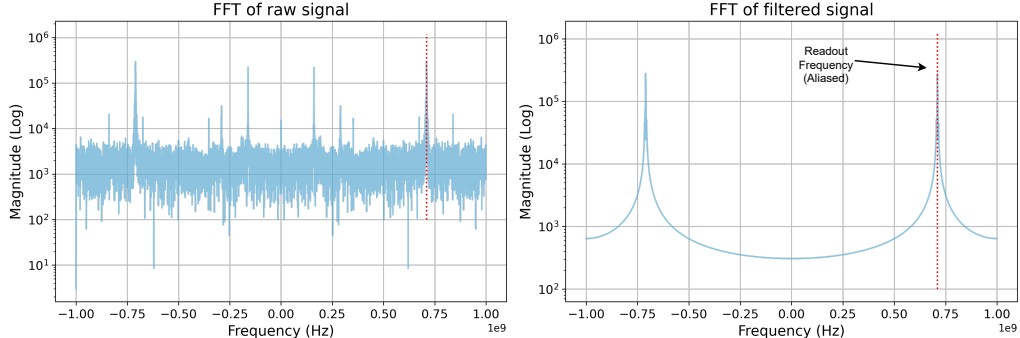

Figure 3: Fourier spectra of the readout signal before and after bandpass filtering. The raw signal (left) exhibits broadband noise across the spectrum, while the filtered signal (right) isolates a sharp peak at the aliased readout frequency, significantly reducing off-resonant noise contributions.

### 3.1.2 BANDPASS FILTERING

In parallel, we employed frequency-domain filtering to suppress noise outside the readout band. Specifically, a $\pm 5$ MHz bandpass filter centered on the qubit readout frequency was applied to each trace. This removes broadband environmental fluctuations while preserving the signal components most strongly coupled to the qubit state. Figure 3 compares the Fourier spectra of raw and filtered signals: the unprocessed data contain strong noise across the band, whereas the filtered spectrum isolates a clean peak at the aliased readout frequency. The corresponding trajectories in Figure 4 show improved clustering and reduced distortion compared to the raw case, further supporting the benefit of filtering.

Together, these pre-processing methods reduce the sensitivity of the classifier to stochastic fluctuations while enhancing the temporal and spectral features that reflect true qubit state dynamics.

### 3.2 LSTM

Time-dependent fluctuations, environmental noise, and relaxation processes can occur at any point during a readout sequence, making classification challenging. Conventional integration-based methods average out these effects, whereas sequence-aware models are capable of learning features distributed across the entire trajectory. To address this, we employ a long short-term memory (LSTM) network (Hochreiter and Schmidhuber, 1997), which is well-suited for capturing temporal dependencies in sequential data.

A practical challenge is that directly fitting the full-resolution traces, which contain up to 2000 timesteps per shot, often leads to poor convergence due to the dominance of redundant information. To mitigate this, we perform binning of the time-series before passing it to the LSTM. Each binned input represents the average of consecutive timesteps, effectively reducing noise while retaining the temporal structure. This binning procedure is applied consistently across all variants of the input data: raw traces, path-transformed signals, and bandpass-filtered signals, ensuring a fair comparison between preprocessing strategies.

The LSTM architecture consists of one or more hidden layers with tunable size, followed by a sigmoid activation layer for multi-class classification. Training is performed using categorical cross-entropy loss. To further improve performance, we adopt a sample-weighting scheme inspired by cost-sensitive learning (Zadrozny et al., 2003). Specifically, points near the centers of Gaussian mixture model (GMM) clusters, where misclassifications are least expected, are weighted more heavily. Intuitively, these central points represent "clean" readouts that are well-separated in the I-Q

plane and correspond to qubits that remained in their prepared states throughout the readout window. By giving them higher weights, the LSTM is penalized more strongly for misclassifying cases that even a simple GMM would handle correctly.

In this way, the weighting scheme strikes a balance: it anchors the classifier to perform reliably on high-confidence data while encouraging it to improve upon the conventional GMM in the more challenging regions of state overlap. Performance across hidden layer sizes and preprocessing configurations is presented in Section 4. As shown there, the combination of LSTMs with binning enables stable training and consistent accuracy improvements over baseline classifiers.

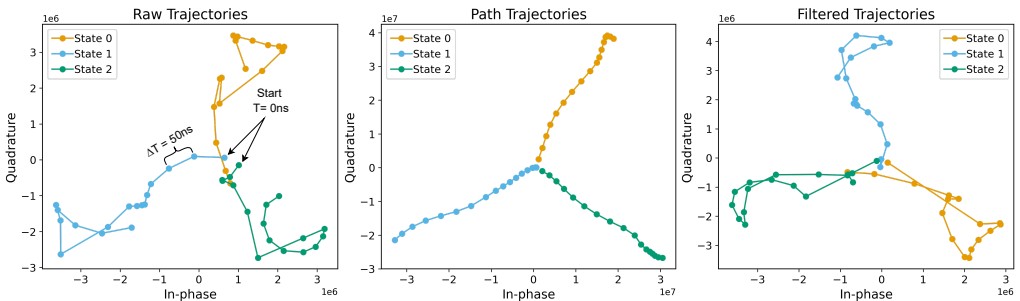

Figure 4: Comparison of qubit state trajectories in the I-Q plane under different preprocessing methods. Raw trajectories (left) contain irregular fluctuations and overlapping regions between states. Path-transformed trajectories (middle) smooth out short-timescale noise and highlight longer-term dynamics, leading to clearer separation between states. Filtered trajectories (right) suppress high-frequency noise, further improving cluster structure and discriminability.

## 4 RESULTS

All experiments were performed with an 80-20 train–test split on datasets collected from real superconducting transmon qubits. Each dataset contained approximately 25,000 shots per state, or 75,000 shots per qubit. Ground truth labels were defined by the control pulses sent to prepare each qubit state. For example, a shot was labeled as state $|1\rangle$ if the board was instructed to apply an X-180 drive pulse, regardless of whether the qubit fully transitioned into $|1\rangle$. The accuracy metric used for comparison is purely based on how many states are read out and classified the same as it was prepared. In a noisy quantum system such as this one, there is no known way to distinguish between all inaccuracies from the physical chip or misclassifications of the points. A qubit can be in a physical state that is different from the prepared one but have the classifier somehow classify it correctly. While this would be problematic for implementing algorithms which require continuous operation on a quantum state, such errors are estimated to be negligibly small after proper heralding and calibration. Therefore significant improvements in classification most likely indicate an improved ability to identify features in noise or decay which would otherwise be lost in the process of integration.

The readout pulses used in our experiments had durations generally in the range of 750 ns to 1000 ns, which is typical for high-fidelity dispersive readout schemes balancing speed and measurement accuracy.

### 4.1 LSTM VS. BASELINE CLASSIFICATION

Across all eight qubits, the LSTM classifier achieved higher readout fidelities than the Gaussian Mixture Model (GMM) baseline (Table 2). Figure 5 illustrates the source of these gains: the LSTM correctly classifies many boundary points between neighboring clusters (panel B), where transient fluctuations or relaxation processes make states ambiguous. Importantly, the LSTM does not misclassify points at the centers of clusters (panel C), ensuring robustness on high-confidence data while improving accuracy in difficult regions. This demonstrates that temporal information preserved in the raw traces provides a genuine advantage over integration-based methods.

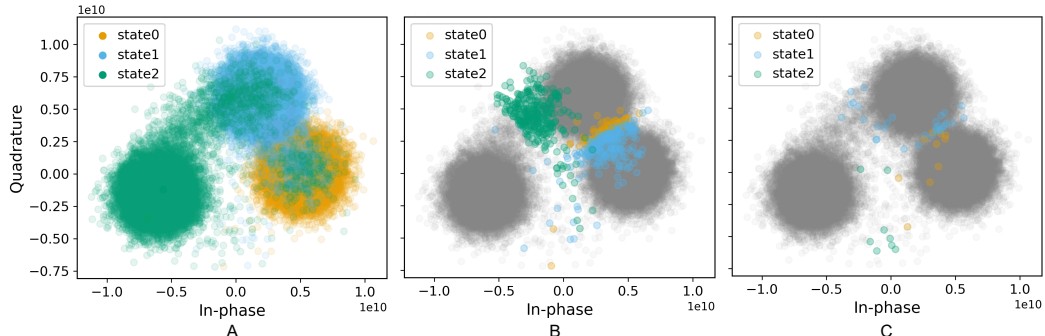

Figure 5: Comparison of LSTM and GMM classification performance in the I-Q plane. (A) Cluster distributions of the three prepared states ($|0\rangle$, $|1\rangle$, $|2\rangle$), showing the overlap regions where misclassifications are most likely. (B) Points highlighted in color indicate cases correctly classified by the LSTM but misclassified by the GMM. These improvements are concentrated near cluster boundaries, where transient fluctuations and relaxation effects create ambiguity. (C) Points highlighted in color represent cases misclassified by the LSTM but correctly classified by the GMM, which are comparatively fewer. Overall, the LSTM provides more robust classification in boundary regions by leveraging temporal information from the full readout trace.

A detailed state-resolved comparison is shown in Table 1, where LSTM + filtering improves GMM performance across nearly all qubits and states, with the most pronounced gains in the $|1\rangle$ subspace.

In summary, LSTM classifiers outperform GMMs by recovering ambiguous boundary cases while maintaining accuracy on well-separated cluster centers.

## 4.2 COMPARISON OF PREPROCESSING METHODS

We next compared LSTM classifiers trained on raw, path-transformed, and filtered signals. In all cases, binning of the input sequences was applied to suppress high-frequency noise and reduce model complexity. As shown in Table 2, performance differences between raw, path, and filtered inputs are small; however, filtering consistently provides the best or near-best fidelity across qubits. Path-based features occasionally smooth trajectories but do not offer significant improvements over raw inputs. In addition, these same improvements are demonstrated with the Dense Neural Network, with an improvement of over .8% on average. We therefore identify **bandpass filtering** as the most reliable approach across devices and across neural network architectures.

| | GMM | | | LSTM+Filter | | |
|---|---|---|---|---|---|---|
| Qubit | $0|0\rangle$ | $1|1\rangle$ | $2|2\rangle$ | $0|0\rangle$ | $1|1\rangle$ | $2|2\rangle$ |
| 0 | 0.988 | 0.972 | 0.962 | 0.996 | 0.990 | 0.961 |
| 1 | 0.994 | 0.983 | 0.973 | 0.997 | 0.985 | 0.978 |
| 2 | 0.989 | 0.975 | 0.958 | 0.996 | 0.978 | 0.967 |
| 3 | 0.954 | 0.879 | 0.971 | 0.972 | 0.940 | 0.970 |
| 4 | 0.988 | 0.930 | 0.951 | 0.996 | 0.946 | 0.956 |
| 5 | 0.999 | 0.976 | 0.964 | 0.998 | 0.985 | 0.969 |
| 6 | 0.991 | 0.864 | 0.938 | 0.991 | 0.912 | 0.935 |
| 7 | 0.996 | 0.944 | 0.902 | 0.998 | 0.962 | 0.900 |

Table 1: State-wise classification fidelity for GMM vs. LSTM+Filter across all qubits.

## 4.3 ERROR MITIGATION AND ROBUSTNESS

The largest accuracy gains appear in the $|1\rangle$ and $|2\rangle$ states, consistent with the fact that these are most affected by noise and fluctuations. Notably, the LSTM also improves $|0\rangle$ classification, highlighting

| Qubit | Baseline (GMM) | LSTM | Path + LSTM | Bandpass filter + LSTM |
|-------|----------------|------|-------------|------------------------|
| Q0 | 0.973 ± 0.002 | 0.980 ± 0.001 | 0.981 ± 0.001 | 0.982 ± 0.001 |
| Q1 | 0.983 ± 0.001 | 0.986 ± 0.001 | 0.985 ± 0.001 | 0.986 ± 0.001 |
| Q2 | 0.973 ± 0.002 | 0.979 ± 0.001 | 0.979 ± 0.001 | 0.980 ± 0.001 |
| Q3 | 0.934 ± 0.002 | 0.961 ± 0.002 | 0.962 ± 0.002 | 0.960 ± 0.002 |
| Q4 | 0.956 ± 0.002 | 0.965 ± 0.002 | 0.965 ± 0.002 | 0.965 ± 0.002 |
| Q5 | 0.979 ± 0.001 | 0.983 ± 0.001 | 0.983 ± 0.001 | 0.983 ± 0.001 |
| Q6 | 0.930 ± 0.002 | 0.944 ± 0.002 | 0.944 ± 0.002 | 0.946 ± 0.002 |
| Q7 | 0.947 ± 0.002 | 0.949 ± 0.002 | 0.940 ± 0.002 | 0.953 ± 0.002 |
| **Avg** | **0.959 ± 0.002** | **0.968 ± 0.002** | **0.967 ± 0.002** | **0.969 ± 0.002** |

| Qubit | DNN | Bandpass filter + DNN |
|-------|-----|-----------------------|
| Q0 | 0.982 ± 0.001 | 0.976 ± 0.001 |
| Q1 | 0.988 ± 0.001 | 0.987 ± 0.001 |
| Q2 | 0.948 ± 0.001 | 0.979 ± 0.001 |
| Q3 | 0.930 ± 0.002 | 0.964 ± 0.002 |
| Q4 | 0.967 ± 0.002 | 0.967 ± 0.002 |
| Q5 | 0.979 ± 0.001 | 0.985 ± 0.001 |
| Q6 | 0.941 ± 0.002 | 0.946 ± 0.002 |
| Q7 | 0.953 ± 0.002 | 0.952 ± 0.002 |
| **Avg** | **0.961 ± 0.002** | **0.969 ± 0.002** |

Table 2: Comparison of average classification fidelity across three states for different models.

its sensitivity to measurement-induced transitions and other non-decay errors. Improvements are observed regardless of qubit coherence times, indicating that the model captures general temporal signatures beyond simple $T_1$ relaxation. This suggests that LSTM-based classifiers remain useful as superconducting hardware advances toward longer coherence and higher fidelities.

### 4.4 IMPLEMENTATION AND TRAINING

All LSTM models were trained with consistent hyperparameters: a learning rate of 0.0001 with exponential decay every 10 epochs, batch size of 256, and 100 epochs. The LSTM model for comparison had 16 nodes. The comparison dense neural network was done with three layers of 32,16,and 8 hidden nodes and the middle layer with a SELU activation and an output layer with softmax activation as utilized in (Lienhard et al., 2022). One of the dense neural networks was done with sample weighting, binning and filtering while the other was done directly on the raw time series data. Binning was critical to ensure convergence: without binning, models required significantly larger hidden layers to beat the GMM baseline, whereas binning allowed smaller models (as few as 4–8 hidden nodes, corresponding to 112–768 parameters) to achieve superior performance. Our compared LSTM model of 16 hidden nodes has 1264 parameters, while the dense neural network we tested against involved 23351 parameters. When both these models were tested for their speed on batch sizes of one, they both recorded similar classification times of 50±10 ms. Therefore, the LSTM was capable of being much more memory efficient in this comparison with far less parameters needed at the same classification speed. **The compactness of the LSTM model make the approach feasible for deployment in real-time readout hardware.**

This consistency in hyperparameters and efficiency of small models highlights the practicality of deploying LSTM classifiers directly in superconducting qubit readout pipelines.

## 5 RELATED WORKS

Time-dependent classifiers for qubit readout have been investigated in several prior studies. Hidden Markov Models (HMMs), deep neural networks, and path signature–based classifiers are among the most common approaches (Martinez et al., 2020; Lienhard et al., 2022; Cao et al., 2025). HMMs are particularly effective at detecting decay events, as they explicitly model state transitions during the readout pulse. However, their improvements are strongly dependent on readout length and qubit coherence time, and they are less effective in addressing other sources of time-dependent noise such as environmental fluctuations.

Neural network–based classifiers, including deep feed-forward networks and autoencoders, have demonstrated fidelity improvements by leveraging nonlinear feature learning. These methods provide advantages such as transfer learning and flexible adaptation to new datasets, but they often require large parameter counts or qubit-specific tuning. Path signature methods [ (Cao et al., 2025)] offer another alternative by compressing trajectories into structured features, which can then be classified by random forests or similar models. While path signatures capture decay-like features effectively, they introduce substantial computational overhead and, at higher orders, require more features than simpler binning strategies without consistently yielding better results.

In comparison, our LSTM-based approach retains the key strengths of time-series models—namely the ability to exploit temporal correlations—while remaining lightweight and hardware-efficient. Across the same datasets, the GMM baseline achieved an average fidelity of about $95.9\%$, while path signature random forest [Cao et al. (2025)] methods improved slightly to $96.2\%$. Our LSTM classifier with bandpass filtering reached $96.9\%$, providing the most consistent improvement across all qubits. Moreover, the model trains with consistent hyperparameters across devices, requires fewer parameters than many deep learning alternatives, and demonstrates robust improvements even on high-coherence, less noisy qubits. This combination of consistency, efficiency, and generalizability makes LSTM + filtering a practical choice for near-term superconducting quantum processors.

## 6 CONCLUSION AND FUTURE WORK

Measurement remains a leading source of error in superconducting quantum processors. While conventional approaches rely on time-integrated classifiers such as Gaussian Mixture Models, we demonstrated that an LSTM-based classifier applied to bandpass-filtered readout traces can better exploit temporal correlations. Using data from eight superconducting transmon qubits, our method achieved an average $\sim 1\%$ fidelity improvement over GMMs, with the greatest gains in the $|1\rangle$ and $|2\rangle$ subspaces. Crucially, the LSTM improved classification of ambiguous boundary cases without sacrificing accuracy on high-confidence points, confirming its robustness to time-dependent fluctuations and noise.

These results highlight LSTMs as a lightweight and hardware-feasible enhancement to readout fidelity, directly contributing to improved circuit performance and reduced overheads for fault tolerance. Future work will extend this approach to multi-qubit readout, where correlated noise and crosstalk present new challenges, and explore integration into FPGA-based hardware for real-time feedback. More generally, sequence models such as LSTMs provide a flexible framework that can be applied across qubit modalities, making time-series–based classification a practical path toward more reliable quantum computing.

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
