# OpenReview forum: "Time-series based quantum state discrimination"
_ICLR.cc/2026/Conference — Submitted to ICLR 2026_

### Official Review · Reviewer_wCGo · 2025-10-28

**Soundness:** 2
**Presentation:** 2
**Contribution:** 3
**Rating:** 6
**Confidence:** 4

**Summary:**

This paper utilizes LSTM networks to read out qutrit states from a superconducting transmon device. In typical setups, Gaussian mixture models, or GMMs, are typically employed for the task of identifying the states of the qutrits after integration is done on the raw signals. In doing so, critical information about the measurements is lost, and it can become difficult to identify states along what would be described as the decision boundary in classical machine learning. By using an LSTM, the authors argue that this important temporal information is captured, improving the classification accuracy of the states and therefore, the effective fidelity of the device. To show this, the authors perform a comparison between the GMM and the LSTM on a standard driving task by comparing the final accuracy of the two models. They find that the LSTM outperforms the GMM.

**Strengths:**

* The problem of error correction is, as clearly stated by the authors, a pressing issue in the community. Further, it appears that the LSTM approach improves the effective fidelity of these devices.
* The experiments were conducted on real quantum devices, removing issues with simulation parameters and noise modeling.
* The concepts are made clear to a broader audience, and in general, the paper is written very well.

**Weaknesses:**

While overall the paper is quite sound and of interest, there were some parts that lacked clarity and accuracy.

* Why do the authors not show the direct classification results? Showing only where one model gets it right but the other wrong does little to highlight the performance of the models. Further, without showing where both models go wrong, it isn't clear whether this comes down to architecture or other factors.  I see that this is included as fidelity scores later in the paper, but perhaps the structure could be cleaned up to show the raw results, followed by how the LSTM beats the GMM.
* The areas where the LSTM beats the GMM are the same as where the GMM beats the LSTM. In the caption of Figure 5, only the former is highlighted as a feature. True, there are more points captured by the LSTM, but I think this needs to be elaborated on. In general, how much of that area is correctly or incorrectly classified? Judging by the fidelity results shown later, it is quite a small amount. That isn't to say it's not impactful, but I think this needs to be made clearer.
* It is not clear that the structure processing of time in the LSTM is the reason for the improvements in the LSTM model. To make this claim, the authors would likely need to perform ablation studies with different context lengths.
* Further, would a dense network with all measurements through time also perform well? For small trajectories, this would be feasible and fast. In general, a broader comparison of machine learning architectures would be helpful rather than jumping straight from GMM to LSTM. This would also align better with the goal of integrating these models onto devices.
* I'm curious about the use of the term fidelity in describing the performance of the classification networks. This is for two reasons. One is purely from an understanding perspective. In the broader ML community, discussing fidelity in the context of classification performance will likely confuse people. The second is more technical. Should this value be referred to as a fidelity? I would consider fidelity to be how well my measurement aligned with my inputs. If I use a classification algorithm to classify my measurements, their performance isn't so much fidelity as simple algorithmic accuracy. Consider the case where a machine has perfect fidelity, but my algorithm is trained to also produce the wrong state. Does this system display a fidelity of 0 or 1? I would be interested in hearing the author's thoughts on this.
* Information about model training is very limited. It would be nice to see the training curves, sizes of networks tested, and their performance, particularly under the motivation of embedding them onto a device.

**Questions:**

* Can the authors quantify the preparation errors mentioned? The fidelities of the devices are already reasonably high, although it is stated that these preparation errors are small compared with noise, it would be good to have a general idea of their size.
* How large were the biggest models used?

---

> ### Author Response · Authors · 2025-11-22
>
> We can include the direct classification results, however a number of misclassified points are impossible to identify, which might be the points in the middle of clusters. How a classifier operates on these is not particularly insightful since these are a problem with hardware rather than readout software.
>
> We can include numbers on by how much the LSTM beats the GMM and in what region.
>
> A dense model also tends to perform much better than any non time-series based models and we may include a comparison if allowed although it might be too significant of a change in the paper.
>
> Fidelity is actually a defined error metric in quantum computing. It covers the probability that a machine produces a quantum state that it is supposed to. In this context, it includes everything from the ability of the hardware to correctly drive a state and to correctly read out a state, since the measurement is what we use to identify a state. Therefore, a machine that is perfect but has an incorrect algorithm would be considered to have 0 fidelity, since we ultimately always identify the wrong quantum state.
>
> Note the fidelities of the devices themselves are relatively high but are difficult to quantify since we can only tell how inaccurate our hardware is by measuring it with our readout scheme. The preparation errors can also be hard to quantify but are small compared to the errors caused by noise during the time of readout or operation since this relies on calibration and sensitivity of room temperature electronics rather than low-coherence superconducting based hardware.
>
> Information about the training, hyperparameter tuning, and size of networks will be included as allowed.

---

> > ### Comment · Reviewer_wCGo · 2025-11-24
> > **Response to authors from Reviewer wCGo**
> >
> > Thank you to the authors for responding to the initial review and addressing the points. Overall, I feel it necessary to stress that the points raised in the review aimed at building out the results in the paper rather than criticising them. Some of the replies by the authors appear to attempt to either appease or directly oppose the suggestions. That is a reasonable response, but it would be more constructive to work cooperatively to get the most out of the work, which I feel is of relevance to the community.
> >
> > * The idea behind the inclusion of the dense model was about a non-time-series model. It is always a good baseline to have. If it shows that no time-series model is required, that is a very useful outcome. It is important to understand where the new approach fits in relation to simpler methods, as they should always be the first approach.
> > * The comment regarding accuracy is a bit confusing. Perhaps the authors could provide more information.
> > * To the fidelity comment, the authors are stating that using this metric, if a quantum computer had 100% fidelity but the result was processed by a poor algorithm it would have a fidelity of 0%. To be clear there, I am referring to a situation where we can read out that quantum device, but the readout is processed by an algorithm, as it is done here. I can see the rational but it is hard to argue that is a good metric. Nonetheless, I understand that the authors are not introducing it themselves. The second point to that comment related to the audience. Classical machine learning researchers, likely interested in this work, might not understand fidelity as a metric. Therefore, for a general audience conference, it is oftentimes useful to include information, perhaps alongside the quantum metrics, that is clearer to that broad audience.

---

> > > ### Author Response · Authors · 2025-12-04
> > > **Edits to resubmission.**
> > >
> > > The fidelity metric and why it is practical in this case is explained in results. To summarize, it is not possibly to properly distinguish between all types of errors, including those created by the classifier, therefore it is common practice to utilize the fidelity of the entire system as the accuracy metric for quantum computers.  In addition, more baseline comparisons are added.

---

### Official Review · Reviewer_LG1n · 2025-10-30

**Soundness:** 2
**Presentation:** 2
**Contribution:** 2
**Rating:** 2
**Confidence:** 2

**Summary:**

The paper is concerned with quantum state discrimination: the problem of accurately reading out quantum states. Typical approaches for this problem employ Gaussian mixture model-based clustering algorithms. Here, the authors suggest to use LSTMs in combination with filtering and feature engineering approaches for quantum state discrimination based on the time series of the readout signal.

**Strengths:**

Currently used methods do not employ temporal information, so employing sequence models for incorporating time series data is well-motivated. Thus, the problem is very relevant for quantum computing, as quantum error correction schemes require high quantum state readout fidelity.

The experiments are based on real data and appear to be pratictically relevant. The experimental results (Table 2) indicate a consistent improvement of LSTM-based methods over the GMM-baseline method.

**Weaknesses:**

My main concern with this paper is that, from a machine learning perspective, the innovation is very limited. LSTMs are standard models for sequence modelling and applying them to time-series classification is well-established. Thus, there do not seem to be any insights for a broader ML audience. The main challenge and contribution of the paper appears to be in the data pre-processing step. Then, any time-series classification method could be applied. Overall, the paper is written in a way that is much more suitable to a quantum computing venue, with the main emphasis put on physical details and on applying off-the-shelf ML methods to quantum state discrimination.

In addition, I also have several concerns regarding presentation and experiments.

Missing experimental details: the paper does not provide sufficient details on the experiments to make them reproducible. For example, the number of layers / hidden nodes in the LSTMs are not specified. The chosen bin size for the binning procedure is not specified. The paper does not appear to use a validation set for selecting these hyper-parameters. Does Table 2 report classification accuracy on the test set?

Methodology: the paper does not clearly explain how the LSTM is trained. The baseline method (GMM clustering) is an unsupervised learning method. In contrast, LSTM-based classification requires labeled training data and it remains unclear where this data is obtained from.

Computing times: The paper states that the LSTMs make the approach feasible for deployment in real-time readout hardware (line 348), but does not report the compute times for the different methods. To support such a claim, also compute times need to be reported.

Role of LSTMs: with the key step being the data preprocessing, the paper does not sufficiently examine the role of LSTMs for the improved accuracies. Does the improved accuracy stem from the improved data preprocessing step? Or does it actually stem from using LSTMs? Insight on this could be gained by including other time-series classification models in the comparison, for example, standard RNNs or state-of-the art time-series classifiers such as
ROCKET: Exceptionally fast and accurate time series classification using random convolutional kernels by Angus Dempster, François Petitjean, Geoffrey I. Webb.

Limited baselines: Since the introduction of GMM methods for this problem in 2014, there have been several other works that employ deep learning-based classifiers for quantum state discrimination (such as B. Lienhard et al. 2022 cited in the paper). It would be important to compare also to such methods.

Wrong key reference: the paper attributes LSTMs to Bengio et al. 2000 (see line 202). However, the original reference is the very well-known paper by Hochreiter and Schmidhuber (1997).

Activation function: the method uses a sigmoid activation function for multi-class classification, which is very uncommon. Rather, softmax activations are typically used, since they restrict the outputs to sum to one. How do you handle cases where you get very low/high probabilities for all cases?

**Questions:**

- How did you select the number of layers / hidden nodes in the LSTMs?
- How did you select the bin size for the binning procedure?
- Does Table 2 report classification accuracy on the test set?
- Can you be more specific on how the labels for training the LSTMs were generated?
- Can you report compute times for your different methods?
- Can you provide further insights on the importance of using LSTMs vs. the improved data pre-processing step?
- Using a sigmoid activation function for a multi-class classification problem, how do you handle cases where you get very low/high probabilities for all cases?

---

> ### Author Response · Authors · 2025-11-22
>
> We submitted this paper to ICLR as an application of deep learning methods to an important field of research.
>
> Certain details will be added in a revision of the paper. This includes
>
> The number of layers and hidden nodes in the LSTM which was one layer with 64 hidden nodes.
>
> The chosen bin size for the comparison, which was bins of 50.
>
> Hyperparameters such as bin size were found by training models of different hidden node counts from 4 - 128 across bin sizes of 5, 10, 20, 50, and 100 on multiple qubits. Training and testing procedure were the same as for the final comparison against the GMM.
>
> Table 2 was tested on a validation set 20% the size of the full data set. This was randomly selected from the full data set.
>
> Labels for the data are based on what signal the controller sends to the qubit. For example, all states labeled one had the controller send a drive pulse to set a qubit in the one state. This means some states will not physically be in the correct state despite being labeled as one, but this number is extremely small. Heralding also mitigates this risk by removing all states which are incorrectly labeled before driving the qubit.  The network was trained with adam optimizer with categorical cross entropy as the loss function.
>
> Compute times will be taken for our models and added.
>
> The LSTM outperformed the baseline models even in the worst case  of data preprocessing, so it has advantages. In addition, preprocessing did not create as significant difference and mostly assisted in reducing complexity so that models could be more efficient and quick.  However, the LSTM has yet to be tested against mentioned time series classifiers on the best case of data preprocessing.
>
> Other baselines that were tested and not included includes random forest classifier on path signature, svm on time-integrated data, FNN on time-integrated data, and FNN on time series data.
>
> The wrong key reference will also be fixed.

---

> ### Comment · Reviewer_LG1n · 2025-11-25
>
> The reply clarifies several points about experimental details. However, the reply only partially address one of the main concerns about the paper: limited experimental validation of LSTMs in comparison to other ML methods.
>
> 1) Authors state: *Other baselines that were tested and not included includes random forest classifier on path signature, svm on time-integrated data, FNN on time-integrated data, and FNN on time series data.* Without reported experimental results, this statement is of very limited use. If future studies want to reproduce this result they will need to know what was the precise experimental setup, which ranges of parameters were tested and what performance was obtained.
> Could you provide at least first experimental results about a comparison to simple RNNs or FNN?
>
> 2) Same comment applies to other baselines from the literature: since the introduction of GMM methods for this problem in 2014, there have been several other works that employ deep learning-based classifiers for quantum state discrimination (such as B. Lienhard et al. 2022 cited in the paper). It would be important to compare also to such methods and/or do a detailed comparison with other ML methods (as mentioned in 1)).
>
>
> 3) From the explanation given, it seems that model hyperparameters were selected on a validation set equal to the **test set**. Authors state: *Hyperparameters such as bin size were found by training models of different hidden node counts from 4 - 128 across bin sizes of 5, 10, 20, 50, and 100 on multiple qubits. Training and testing procedure were the same as for the final comparison against the GMM.* and *Table 2 was tested on a validation set 20% the size of the full data set. This was randomly selected from the full data set.* Main text also only mentions that a *80-20 train–test split* was performed. This indicates that LSTM hyperparameters were selected based on performance on the test set, rather than a separate validation set.
>
> 4) Compute times: can you provide a first rough indication of how the methods compare in terms of speed?

---

> > ### Author Response · Authors · 2025-12-04
> > **New additions to resubmission**
> >
> > Added the FNN comparison. Simple RNN comparison is mentioned, although it is generally the worst performing of the three architectures. In addition the model used in the Lienhard paper, three layer FNN with no sample weights or preprocessing is compared. Rough estimate of the compute time is also given for inference.

---

### Official Review · Reviewer_v6GT · 2025-10-31

**Soundness:** 3
**Presentation:** 3
**Contribution:** 2
**Rating:** 4
**Confidence:** 2

**Summary:**

In this paper, a novel approach to efficiently and accurately readout quantum states is presented. This task is curcial for quantum error correction.

The authors proposed to use machine learning for this task. In particular a LSTM model, in bomcination with filtering and feature engineering techniques is presented.

The proposed approach outperform clustering models and is better that the proposed time-series baseline (GMM).

The application of an LSTM-based classifier to bandpass-filtered readout traces is not well aligned with ICLR’s core focus on methodological advances in machine learning.

**Strengths:**

- Readout fidelity is a well-known bottleneck in superconducting qubit systems
- Time-series framing of qubit readout is promising
- The proposed approach is novel
- Quality of the presentation is high

**Weaknesses:**

- The reported improvement is small, even if significant
- Only GMM is used for comparison
- The topic is not closely aligned with ICLR’s core areas of interest

**Questions:**

Why do you compare only against GMMs? Are there other time-series ML models that could be used for this purpose?

---

> ### Author Response · Authors · 2025-11-21
>
> At the high accuracy rates that current systems are at, even small improvements are valuable. As qubits gets less noisy, the amount of points which any classifier might find difficult to classify decreases. Therefore, improvements will be less noticeable. Notably, the biggest improvements occur when the qubit is especially noisy like in qubit 3. Still, hardware is expected to be quite noisy for the near-future so our improvements still have value for quantum computing readouts. Additionally,  significant improvements are also not possible for certain qubits, as the hardware is at fault in many cases.  Most alternative improvements come from upgrading hardware which is more resource intensive and not generalizable across many systems like our software. When standard error of the model is taken, it still outperforms the GMM in all cases.
>
> As for comparison to other models, we did test it against other models including time series based ones including a random forest classifier on a path signature and dense neural networks on integrated or time series data. Our LSTM still had significant improvements over most of the aforementioned comparisons except the dense neural network on time series data. The dense neural network often matched the LSTM performance except on certain qubits, notably qubit 3.
>
> We thought this research paper might align with ICLR as it falls under an application of deep learning method to an important area of research.

---

### Meta-Review · Area_Chair_VWUa · 2026-01-06

**Summary:**

The reviewers acknowledge the practical relevance of the problem (improving quantum state readout fidelity) and the potential of using time-series data. However, there is a consensus that the paper lacks sufficient methodological innovation for ICLR, as it applies standard off-the-shelf LSTMs. Major concerns include the lack of comprehensive baselines (initially only comparing to GMMs, missing other deep learning classifiers or simple RNNs/FNNs), insufficient ablation studies to distinguish the benefits of preprocessing versus the model architecture, and missing experimental details regarding training and validation.

**Reviewer Concerns:**

**Addressed:**
The authors clarified specific experimental details such as the number of layers, hidden nodes, and bin sizes. They also acknowledged the incorrect citation for LSTMs and provided a preliminary comparison with an FNN in the discussion phase.

**Outstanding:**
Reviewer **LG1n** and **v6GT** remain concerned about the limited novelty from a machine learning perspective. The comparison against relevant baselines (e.g., state-of-the-art time-series classifiers, other deep learning quantum discrimination methods like Lienhard et al.) remains insufficient or merely anecdotal without rigorous data in the paper. Reviewer **wCGo** and **LG1n** noted that the analysis does not clearly attribute the performance gains to the LSTM architecture versus the data preprocessing, as ablation studies are missing.

**Reviewer Scores:**

- Reviewer v6GT: 4. The reviewer views the paper as an application of standard methods with limited methodological alignment for ICLR. They also criticized the lack of comparison against other time-series models beyond GMMs.

- Reviewer LG1n: 2. The reviewer sees minimal ML innovation and argues that the experimental validation is flawed due to missing baselines and details. The rebuttal failed to provide concrete evidence that LSTMs are superior to standard RNNs or FNNs through rigorous testing.

- Reviewer wCGo: 6. The reviewer acknowledges the practical value on real hardware but questions the necessity of the LSTM architecture. They pointed out the lack of ablation studies to distinguish the model's contribution from data preprocessing or simpler dense networks.

---

### Decision · Program_Chairs · 2026-01-26

Reject